# Self-management for chronic widespread pain including fibromyalgia: A systematic review and meta-analysis

Adam W. A. Geraghty[1]*, Emma Maund[1], David Newell[1,2], Miriam Santer[1], Hazel Everitt[1], Cathy Price[3], Tamar Pincus[4], Michael Moore[1], Paul Little[1], Rachel West[1], Beth Stuart[1]

1 Primary Care Research Centre, School of Primary Care, Population Sciences and Medical Education [PPM], Faculty of Medicine, University of Southampton, Southampton, United Kingdom, 2 AECC University College, Bournemouth, United Kingdom, 3 Solent NHS Trust, Southampton, United Kingdom, 4 Department of Psychology, Royal Holloway, University of London, London, United Kingdom

* A.W.Geraghty@soton.ac.uk

## Abstract

### Background

Chronic widespread pain (CWP) including fibromyalgia has a prevalence of up to 15% and is associated with substantial morbidity. Supporting psychosocial and behavioural self-management is increasingly important for CWP, as pharmacological interventions show limited benefit. We systematically reviewed the effectiveness of interventions applying self-management principles for CWP including fibromyalgia.

### Methods

MEDLINE, Embase, PsycINFO, The Cochrane Central Register of Controlled Trials and the WHO International Clinical Trials Registry were searched for studies reporting randomised controlled trials of interventions adhering to self-management principles for CWP including fibromyalgia. Primary outcomes included physical function and pain intensity. Where data were sufficient, meta-analysis was conducted using a random effects model. Studies were narratively reviewed where meta-analysis could not be conducted Evidence quality was rated using GRADE (Grading of Recommendations, Assessment, Development and Evaluations) (PROSPERO-CRD42018099212).

### Results

Thirty-nine completed studies were included. Despite some variability in studies narratively reviewed, in studies meta-analysed self-management interventions improved physical function in the short-term, post-treatment to 3 months (SMD 0.42, 95% CI 0.20, 0.64) and long-term, post 6 months (SMD 0.36, 95% CI 0.20, 0.53), compared to no treatment/usual care controls. Studies reporting on pain narratively had greater variability, however, those studies meta-analysed showed self-management interventions reduced pain in the short-term (SMD -0.49, 95% CI -0.70, -0.27) and long-term (SMD -0.38, 95% CI -0.58, -0.19) compared

**Data Availability Statement:** All relevant data are within the paper and its Supporting Information files.

**Funding:** This research was part funded by a National Institute for Health Research (NIHR) Research Capacity Funding Award from Solent NHS Trust to Dr Emma Maund. The NIHR and Solent NHS Trust had no involvement in the design of the review protocol, and had no input in data collection, analyses, interpretation, writing up or publication of the review.

**Competing interests:** The authors have declared that no competing interests exist.

to no treatment/usual care. There were few differences in physical function and pain when self-management interventions were compared to active interventions. The quality of the evidence was rated as low.

## Conclusion

Reviewed studies suggest self-management interventions can be effective in improving physical function and reducing pain in the short and long-term for CWP including fibromyalgia. However, the quality of evidence was low. Future research should address quality issues whilst making greater use of theory and patient involvement to understand reported variability.

## Introduction

Chronic widespread pain (CWP) has a reported prevalence of between 9.6%-15% in the general population [1, 2] and is diagnosed when long-lasting pain occurs across multiple body sites [2]. CWP is the defining feature of fibromyalgia, where widespread pain is accompanied by fatigue, waking unrefreshed and cognitive symptoms [3]. Fibromyalgia has a reported global prevalence of 2.7% [4] and is increasingly viewed as representing the severe end of a CWP spectrum [2, 5, 6]. Guidelines for the management of CWP including fibromyalgia recommend non-pharmacological interventions as first-line care [6–8], with a limited number of pharmacotherapies used to manage severe symptoms [e.g. pain, sleep problems]. The reduced focus on pharmacological management in CWP [7, 8] aligns with the increasing importance placed on psychosocial and behavioural self-management for this complex pain condition [3, 6, 9].

Self-management refers to an individual's ability to monitor their health condition and effect the behavioural, cognitive and emotional responses required to support a satisfactory quality of life [10]. Definitions vary with regard to details of specific skills necessary for self-management. However, there is broad consensus that in self-managing, individuals are active in developing, applying and maintaining appropriate skills in their day-to-day lives [10–12]. Additionally, self-management reflects a multidimensional process [10, 11, 13]. Support for self-management should thus cover multiple domains; providing the greatest opportunity for individuals to gain the understanding necessary to appropriately regulate the behavioural, cognitive and affective impacts of chronic illness.

Within the CWP review literature, the majority of systematic reviews have combined all non-pharmacological interventions [14, 15], or focused on single non-pharmacological treatment approaches including exercise [16], CBT [17] and mindfulness [18]. Häuser et al.'s [19] multicomponent therapy review is the closest to aligning with principles of self-management described above. In their 2009 review of 9 randomised controlled trials (RCTs), multicomponent therapy was defined as an intervention that had an exercise component combined with an educational/psychological component [19]. Häuser et al. found these combined interventions had beneficial short-term effects for fibromyalgia, but longer-term effects were limited [19].

As support for self-management is increasingly called for by both patients and clinicians [6, 20], understanding the broad effectiveness literature in this domain is critical to support the implementation and development of effective self-management interventions for CWP. For the current review, we drew on a definition of self-management that aligned conceptually with

key aspects of a self-management approach [10–13]; combining multiple components and the teaching of skills that could be applied beyond the intervention: Miles et al. [21] define self-management interventions as multicomponent programmes which aim to improve health or quality of life, with opportunities for improvements in individuals' abilities to manage their own health. They should also aim to increase skills and knowledge, enabling individuals to use these skills beyond intervention periods [21]. This definition has been applied across a number of pain-related reviews of self-management interventions [21]. Thus, our primary aim was to build on the earlier work of Häuser et al. [19], by using this specific definition of a self-management approach, and systematically reviewing the effectiveness of interventions applying these self-management principles to CWP including fibromyalgia. As a secondary aim, we wished to explore the impact of delivery modality on effectiveness (e.g. group format vs. internet delivery).

## Methods

### Protocol and registration

A protocol was developed (see S1 File) and used for as the basis of preregistration on PROSPERO (https://www.crd.york.ac.uk/prospero/), reference number: CRD42018099212.

### Information sources/search strategy

We searched the following electronic databases: Cochrane Central Register of Controlled Trials (CENTRAL), MEDLINE (Ovid), Embase (Ovid), PsycINFO (EBSCO*host*), and the WHO International Clinical Trials Registry platform. Searches were conducted from inception to December 2017 and updated from inception to June 2020. In the updated search, the WHO International Clinical Trials Registry platform could not be accessed due to COVID-19 restrictions. The primary searches were supplemented with reference list checking. Database specific search strategies were developed using subject headings and text words related to CWP, fibromyalgia, self-management interventions, and database specific RCT filters. The MEDLINE strategy was developed first and reviewed by a Medical Librarian. After the MEDLINE strategy was finalised (see S2 File for the MEDLINE strategy), it was adapted to the syntax and subject headings of the other databases. There were no language restrictions applied. All searches were conducted by systematic review specialist, EM.

### Eligibility criteria

We included studies employing a randomised controlled trial (RCT) design recruiting adults aged 18 or over with a diagnosis of CWP or fibromyalgia. Studies where participants had mixed diagnoses (e.g. arthritis and fibromyalgia) were included if data were reported separately for those with CWP/fibromyalgia. We used Miles et al.'s [21] definition of self-management as our criteria for intervention inclusion. Interventions were included if they:

a. Aimed to improve participants' health status or quality of life with opportunity or improvement in participants managing their own health

b. Aimed to increase skills and knowledge of the participants and enable participants to use these skills in their lives beyond the intervention

c. Were directed at patients

d. Were multicomponent, e.g. included exercise and a psychological component. Trials focusing on only single component interventions (e.g. exercise or relaxation alone) were excluded.

Following Miles et al.'s [21] definition, to be considered self-management, the intervention had be comprised of at least two components from the following five: psychological (including behavioural or cognitive therapy, or an alternative approach that taught skills), mind-body therapies (MBT) (including components such as relaxation, meditation or guided imagery), physical activity (any form of exercise), lifestyle (such as dietary advice and sleep management) and medical education (such as information to support patients' understanding of their condition and effective use of medication).

We included trials where a self-management intervention was compared to a range of comparators including waiting lists and treatment as usual, alternative interventions (e.g. single component such as exercise alone) and attention controls (where the intention was to control for placebogenic factors).

## Outcomes

Physical function and pain intensity were included as primary outcomes. Both physical function and pain intensity are recommended core outcome domains in chronic pain trials [22] and in CWP studies more specifically [23]. The following secondary outcomes were included: disease specific measures (e.g. Fibromyalgia Impact Questionnaire, FIQ); global health measures; quality of life; mental health [e.g. depression/anxiety, psychological well-being]; harms; medication usage; and healthcare utilisation. We selected the measure deemed most appropriate for each outcome from each included trial [17]. When there was more than one outcome measure for a particular outcome of interest included in a trial, preference was given to the measure most frequently used [17, 24]. We included short-term and long-term data on these outcomes. Short-term was defined as post-treatment to three months, with post-treatment data taking preference. Long-term was defined as follow up at six months or longer. Where more than one outcome occurred at 6 months and beyond, data for the final follow-up were included [17].

## Study selection, data extraction

Two authors independently screened all titles and abstracts yielded by the searches against the inclusion criteria (AG, EM first search, AG, DN updated search). Full text papers were sought for all titles and abstracts that appeared to meet the inclusion criteria or where there was uncertainty. Two reviewers (AG, EM first search, AG, DN updated search) then independently assessed whether these full papers met the inclusion criteria. Disagreements were resolved through discussion, and where applicable, arbitration by a third author (BS). All records identified were considered at the level of studies, consequently we extracted data from included RCT papers presenting results, protocol papers, abstracts and registry entries.

Data were extracted into a pre-piloted Excel data extraction form. Data extracted included: patient characteristics (e.g. age, sex, diagnostic criteria used, duration of illness); elements of the Template for Intervention and Replication ([TIDierR] checklist [25]); and all necessary quantitative data for planned analysis. Data were extracted by an extraction team comprised of EM, AG, RW and DN. Study details were independently double checked by a member of the extraction team who did not perform initial extraction. All extracted quantitative data for analysis was independently double checked by medical statistician BS.

## Risk of bias

Risk of bias assessment was performed by authors from the extraction team (EM, AG, RW, DN) and independently double checked by a member of the team not involved with the primary assessment. Any disagreements were resolved through discussion. We used the

Cochrane Risk of Bias tool [26]. For each included study, following the Risk Bias Tool, we assessed random sequence generation; allocation concealment; blinding of participants and personnel; blinding of outcome assessment; incomplete outcome data; and selective outcome reporting.

Some of the criteria were further operationalised for consistency, in particular 'incomplete outcome data' was assessed in the following way: First, we drew on Detry et al.'s [27] definition of intention to treat (ITT): "Under ITT, study participants are analyzed as members of the treatment group to which they were randomized regardless of their adherence to, or whether they received, the intended treatment" pg 85. If follow-up was above 75% and the ITT principle was followed in the analysis that was classed as low risk. If follow-up was above 75% and it was not clear whether the ITT principle had been followed that equated to unclear risk. If follow-up was below 75% and/or only a per protocol analysis was reported (|e.g. just participants that completed certain aspects of the protocol analysed) was classed as high risk. Following recent work on outcome reporting bias [28], we operationalised the 'selective outcome reporting' criterion in the following way: If a registration document was found, and the outcomes matched the published paper this was classed as low risk. If no registration entry/document was found, we classed this as unclear risk. If a registration document was found with differing outcomes reported to those in the published paper, this was classed as high risk.

## Quality rating

We used the GRADE (Grading of Recommendations Assessment, Development, and Evaluation) approach to rate the quality of evidence in the review for our primary outcomes [29]. When using GRADE, evidence on outcomes from RCTs starts as high quality, and reviewers then rate down for limitations (risk of bias), inconsistency, indirectness, imprecision, and publication bias (ratings range from 'high', to 'very low'). Evidence for outcomes can be rated up considering factors such as very large effects and evidence of dose response gradients [29]. Whilst this approach is frequently coupled with meta-analysis, Murad et al., [30] show how it can also be used in absence of a quantitative estimate of effect. As such, we included both meta-analysed and narratively reviewed studies when grading the quality of evidence for the primary outcomes. Where estimates and confidence intervals were absent in narratively reviewed studies, we took a cautionary approach and rated down for imprecision. We prepared evidence profiles and summary of findings tables for our two comparisons: self-management vs. usual care/no treatment, and self-management vs. active comparison. GRADE ratings were agreed through consensus by a sub-team comprising AG, BS, CP and EM.

## Data analysis and synthesis

Key study characteristics were summarised narratively through text and in tables presenting study aspects and intervention components. Studies including comparisons of self-management interventions vs. no-treatment/waiting list/usual care controls were analysed separately from comparisons of self-management interventions vs. alternative active conditions. All studies meeting our inclusion criteria were judged as similar enough to be entered into meta-analysis if data allowed. If effectiveness data presented in a study were not sufficient for meta-analysis we a used a narrative approach to describe findings pertaining to the relevant included outcomes. We did not attempt to contact study authors for this data, primarily due to the large number of studies falling into this category, many of which were >10 years old; we applied a narrative approach to all for consistency. Meta-analysis was undertaken using a random effects model with Review Manager Software (RevMan 5.4). For continuous outcomes, we presented mean differences with 95% confidence intervals (CIs) where the same measurement scales

were used in all papers, or standardised mean differences (SMD) with 95% CIs where different measurement scales were used. Statistical heterogeneity was tested using the $Chi^2$ test (significance level: 0.1) and $I^2$ statistic. Data for cluster randomised trials were treated according to the methods described in the Cochrane Handbook for Systematic Reviews of Interventions [26], i.e. the total sample size in each arm was adjusted for the design effect. Where at least 10 studies were included in an analysis, we assessed publication bias examining funnel plot asymmetry and applying Egger's test (significance = $p<0.05$). This was done using STATA (version 16.0).

## Results

### Search

The combined first and updated electronic database search resulted in 6,322 records identified once duplicates had been removed (see Fig 1 for full details of studies selection). Following screening, 193 full text articles were assessed for eligibility. One hundred and fourteen articles were excluded based on our eligibility criteria. This resulted in 54 unique studies included in the review, comprising 39 completed trials (36 full publications, and three conference abstracts with results), four protocol papers and 11 clinical trial registry entries. One study was published in Spanish, and one paper in Portuguese. All other included studies were published in English. See S3 File for a full table of characteristics of studies included.

### Participants

The studies reviewed included 6072 participants. The mean age of participants was 48.7 years, and 93.7% were female. The mean duration of time since diagnosis was 8.6 years.

Three studies used chronic widespread pain as their primary classification for participants [31–33]: Two studies used the American College of Rheumatology (ACR) 1990 definition of widespread pain as a diagnosis [34]. One study specified referral from a medical specialist or general practitioner with chronic widespread pain, with or without a diagnosis of fibromyalgia [32]. One study referred explicitly to fibromyalgia and chronic widespread pain; describing that patients were recruited from primary healthcare centres by searching patient journals for the diagnosis of fibromyalgia and chronic widespread pain [35]. All remaining studies used fibromyalgia as a primary classification: One study reported use of Yunus' criteria for fibromyalgia [36], one study referred to the ICD-10 classification for fibromyalgia (M79.7) [37]. Three studies reported a diagnosis of fibromyalgia, but did not specify criteria used [38–40]. All other studies referred to the ACR criteria of fibromyalgia (1990, 2010). Broadly, 46.2% of participants were recruited from rheumatology clinics or hospital specialist settings; 12.8% were recruited from primary care; 15.4% were recruited from a mixture of primary care and rheumatology clinics; 12.8% were recruited from community advertisements, and in a further 12.8% of cases it was not clear where participants were recruited from.

### Delivery modality

Health professional-led groups were the predominant format for delivery, used in 34/36 (94.4%) of the studies that directly reported how delivery was implemented. Health professionals included physiotherapists, psychologists, rheumatologists, general practitioners, nurses and social workers. Three abstract-only studies did not explicitly state how the intervention was delivered [41–43]. In two studies, group sessions were part of inpatient programmes of one week [44] and four weeks [32]. The remainder of the group interventions were outpatient, delivered in timeframes ranging from daily sessions over two weeks [31] to weekly sessions over 21 weeks [45]. One intervention was delivered with a combination of group sessions for

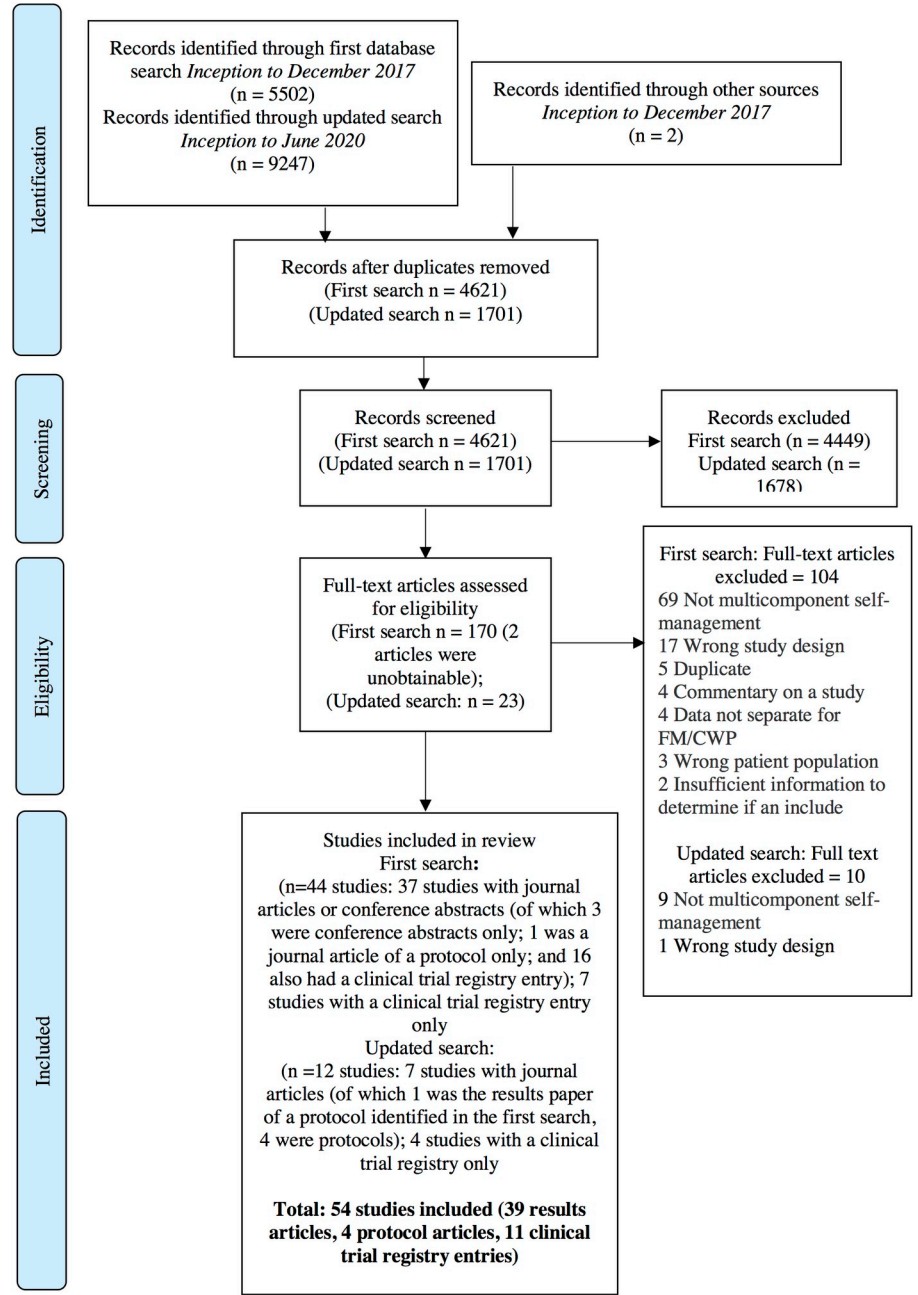

**Fig 1. Study selection flow diagram.**

exercise and individual telephone sessions for CBT [33]. A further intervention combined group inpatient sessions with a follow-on smartphone programme that provided daily individualised feedback from a therapist [32]. Two interventions were delivered using internet-based systems; one internet intervention used moderated online workshops and a moderated bulletin board [38], the other internet intervention was delivered as a stand-alone website [46]. Due to the lack of variability i.e. just two studies that used formats other than groups, it was not possible to explore whether delivery modality was related to effectiveness. Consequently, this analysis was not taken beyond description.

## Intervention content

All included studies reported on interventions that combined at least two of the five components constituting the definition of self-management for this review [21], see S3 File for brief description and Table 1 component breakdown details. The two most common components were physical activity, used in 33/39 (84.6%) of interventions; and medical education, also used in 33/39 (84.6%) of interventions. Physical activity primarily comprised exercise sessions including a focus on aerobic, flexibility and strengthening exercises led by specialists (e.g. physiotherapists). This also included exercise sessions in pools in a smaller number of studies [32, 35, 42, 44, 47–50]. In one study the physical activity component comprised of Qigong, (a gentle movement system grounded in Traditional Chinese Medicine) [51]. In another study Tai Chi was integrated with stretching and flexibility exercises [52]. Medical education primarily comprised of information regarding the medical understanding of CWP/ fibromyalgia, symptoms and treatment options (including pharmacological treatments). Psychological approaches were used in 22/39 (56.4%) of interventions. The approaches were predominately cognitive and/or behavioural, ranging from full Cognitive Behaviour Therapy (CBT) protocols [53], to integration of specific cognitive and behavioural strategies into multidisciplinary programmes [52]. Two studies incorporated different approaches including a technique based on interaction and communication theory [39]; and an expanded autogenic training with a focus on how conflictual emotional experience may reside in the body [54]. Mind-body components were used in 14/39 (35.8%) interventions. The mind-body components comprised of mindfulness [51, 55], meditation [56] and relaxation training [32, 38, 44, 45, 48, 57–60]. Finally, 6/39 (15.4%) reported lifestyle components in interventions, this included a focus on nutrition [44, 56, 61], regulation and adjustment of everyday lifestyle [44, 61, 62], and sleep hygiene [37, 63].

Thirteen studies (33%) described theory underlying the interventions. See S3 File for further description of theories described. Theory relating to cognitive behavioural principles for the remediation of pain was most common [44, 46, 52, 63, 64]. Social cognitive theory, and self-efficacy theory specifically, was mentioned in four studies [44, 52, 61, 62]. Theories relating to mindfulness, awareness and interrelation were described in four studies [39, 45, 51, 55]. Stuifbergen et al. [61] provided a specific logic model/proposed theory of change for their intervention. Kristjansdottir et al. [32] described their intervention as building on a range of theoretical models including cognitive behavioural theories of catastrophising, acceptance and commitment-based theory, and self-determination theory. Martin et al. [65] briefly mentioned the biopsychosocial model as underlying their intervention approach.

## Comparisons

Studies used a range of comparisons (see S3 File for full details). Self-management interventions were compared to a single wait list control, no treatment control or usual care control in 23/39 59% of studies. Three studied referred to attention controls, where the aim of the intervention was to control for 'non-specific' factors likely to be therapeutic [57, 61, 63]: Eight studies compared the focal self-management intervention to one other alternative intervention [32, 35, 41, 45, 51, 52, 60, 66]. Three studies compared the self-management intervention to an additional active intervention and a control group [58, 60, 67]. Four studies compared self-management to multiple interventions components including controls [33, 57, 62, 68].

## Risk of bias

Many studies did not provide enough detail to enable a clear judgment of high or low risk on the risk of bias criteria. See Fig 2 for an overview and Fig 3 for detailed ratings for each study.

**Table 1. Components of interventions.**

| First author | Psychological | Mind–body therapies | Physical activity | Lifestyle | Medical education | Component summary |
|---|---|---|---|---|---|---|
| Amris 2014 | ✓ | | ✓ | | ✓ | Psychological; Physical activity; Medical education |
| Astin 2003 | | ✓ | ✓ | | | Mind-body therapy; Physical activity |
| Bosch 2002 | | ✓ | ✓ | | ✓ | Mind-body therapy; Physical activity; Medical education |
| Bourgault 2015 | ✓ | | ✓ | | ✓ | Psychological; Physical activity; Medical education |
| Buckelew 1998 | | ✓ | ✓ | | | Mind-body therapy; Physical activity |
| Burckhardt 1994 | | ✓ | ✓ | | ✓ | Mind-body therapy; Physical activity; Medical education |
| Castel 2013 | ✓ | | ✓ | | | Psychological; Physical activity. |
| Cedraschi 2004 | | ✓ | ✓ | | ✓ | Mind-body therapy; Physical Activity; Medical education |
| De Souza 2008 | ✓ | ✓ | ✓ | | ✓ | Psychological; Mind-body therapy; Physical activity; Medical education |
| Giannotti 2014 | | | ✓ | | ✓ | Physical activity; Medical education |
| Gowans 1999 | | | ✓ | | ✓ | Physical activity; Medical Education |
| Hammond 2006 | ✓ | | ✓ | | ✓ | Psychological; Physical activity; Medical education |
| Hamnes 2012 | ✓ | ✓ | ✓ | ✓ | ✓ | Psychological; Mind-body therapy; Physical activity; Lifestyle; Medical education. |
| Hsu 2010 | ✓ | ✓ | ✓ | | ✓ | Psychological; Mind-body therapy; Physical activity; Medical education |
| Kendall 2000 | | ✓ | ✓ | | ✓ | Mind-body therapy; Physical activity; Medical education |
| King 2002 | | | ✓ | ✓ | ✓ | Physical activity; Lifestyle; Medical education |
| Koulil 2010 | ✓ | | ✓ | | | Psychological; Physical activity. |
| Kristjansdottir 2013 | ✓ | ✓ | ✓ | | ✓ | Psychological; Physical activity; Lifestyle; Medical education |
| Kubra 2013 | | | ✓ | | ✓ | Physical activity; Medical education |
| Lemstra 2005 | ✓ | ✓ | ✓ | | ✓ | Psychological; Mind-body therapy; Physical activity; Medical education |
| Lera 2009 | ✓ | | ✓ | | ✓ | Psychological; Physical activity; Medical education |
| Lorig 2008 | ✓ | ✓ | ✓ | | ✓ | Psychological; Physical activity; Medical education |
| Luciano 2011 | ✓ | | | | ✓ | Psychological; Medical education |
| Mannerkopi 2000 | | | ✓ | | ✓ | Physical activity; Medical Education |
| Mannerkopi 2009 | | | ✓ | | ✓ | Physical activity; Medical Education |
| Martin 2014 | ✓ | | ✓ | | ✓ | Psychological; Physical activity; Medical Education |
| McBeth 2012 | ✓ | | ✓ | | | Psychological; Physical activity |
| McVeigh 2006 | | | ✓ | | ✓ | Physical activity; Medical Education |
| Rooks 2007 | | | ✓ | | ✓ | Physical activity; Medical education |
| Salaffi 2015 | | | ✓ | | ✓ | Physical activity; Medical education |
| Saral 2016 | ✓ | | ✓ | | ✓ | Psychological; Physical activity; Medical education |
| Stuifbergen | | | ✓ | ✓ | ✓ | Physical activity; Lifestyle; Medical education |
| Tousignant Laflamme 2014 | ✓ | | ✓ | | ✓ | Psychological; Physical activity; Medical education |
| Traistru 2015 | | ✓ | ✓ | | | Psychological; Physical activity |
| Vlaeyen 1996 | ✓ | | | | ✓ | Psychological; Medical education |
| Williams 2010 | ✓ | | | ✓ | ✓ | Psychological; Lifestyle; Medical education |
| Musekamp 2019 | ✓ | | | ✓ | ✓ | Psychological; Lifestyle; Medical education |
| Pe´rez-Aranda 2019 | | ✓ | | | ✓ | Mind-body therapy; Medical education |
| Pereira Pernambuco 2018 | ✓ | ✓ | ✓ | ✓ | ✓ | Psychological; Mind-body therapy; Physical activity; Lifestyle; Medical education. |

*(Continued)*

**Table 1.** (Continued)

| First author | Psychological | Mind–body therapies | Physical activity | Lifestyle | Medical education | Component summary |
|---|---|---|---|---|---|---|
| Araya-Quintanilla 2020 [protocol] | ✓ | | ✓ | | ✓ | Psychological; Physical activity; Medical education |
| Caballol Angelats 2019 [protocol] | ✓ | | ✓ | | ✓ | Psychological; Physical activity; Medical education |
| Haugmark 2018 [protocol] | ✓ | ✓ | | | | Psychological; Mind-body therapies |
| Serrat 2020 [protocol] | ✓ | | ✓ | | ✓ | Psychological; Physical activity; Medical education |
| NCT00715195 | ✓ | | ✓ | | | Psychological; Physical activity |
| NCT00088777 | ✓ | | ✓ | | | Psychological; Physical activity |
| NCT00925431 | | | | ✓ | ✓ | Lifestyle; Medical education |
| NCT03044067 | | | ✓ | | ✓ | Physical activity; Medical education |
| NCT00000398 | ✓ | | ✓ | | | Psychological; Physical activity |
| NCT03641495 | | | ✓ | | ✓ | Physical activity; Medical education |
| ISRCTN96836577 | | ✓ | ✓ | | | Mind-body therapy; Physical activity |
| NCT03073642 | | | ✓ | | ✓ | Physical activity; Medical education |
| ISRCTN10824225 | | ✓ | ✓ | ✓ | ✓ | Physical activity; Mind-body therapy; Lifestyle; Medical education |
| NCT04100538 | ✓ | | ✓ | | ✓ | Psychological; Physical activity; Medical education |
| NCT04220567 | | | ✓ | | ✓ | Physical activity; Medical education |

Note. Gray cells are protocols or trial registry entries.

**Randomisation.** Twenty-four studies reported their randomisation sequence generation procedure in enough detail to be scored as low risk. Fifteen studies did not provide enough detail and were rated as unclear risk. No studies were rated as high risk.

Allocation concealment: Reported allocation concealment procedures were rated as low risk in 19 studies. Eighteen studies did not provide enough detail and were rated as unclear. Two studies were rated as high risk, both stating that it was not possible to blind allocation procedures [37, 64].

**Blinding of participants and personnel.** It is not possible to fully blind participants or the delivery of psychosocial/behavioural interventions. Consequently, all trials were scored as high risk on this criterion.

**Blinding of outcome assessors.** Eighteen trials were scored as low risk reporting that outcome assessors were blinded to allocation. Eighteen trials did not provide enough information

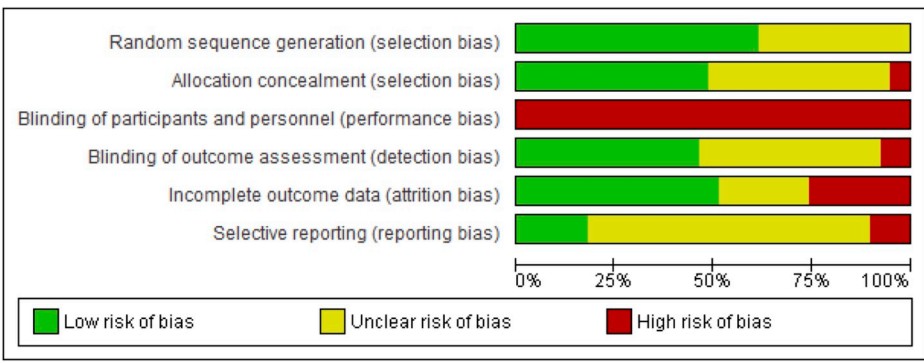

**Fig 2. Overview of risk of bias scorings for included completed studies.**

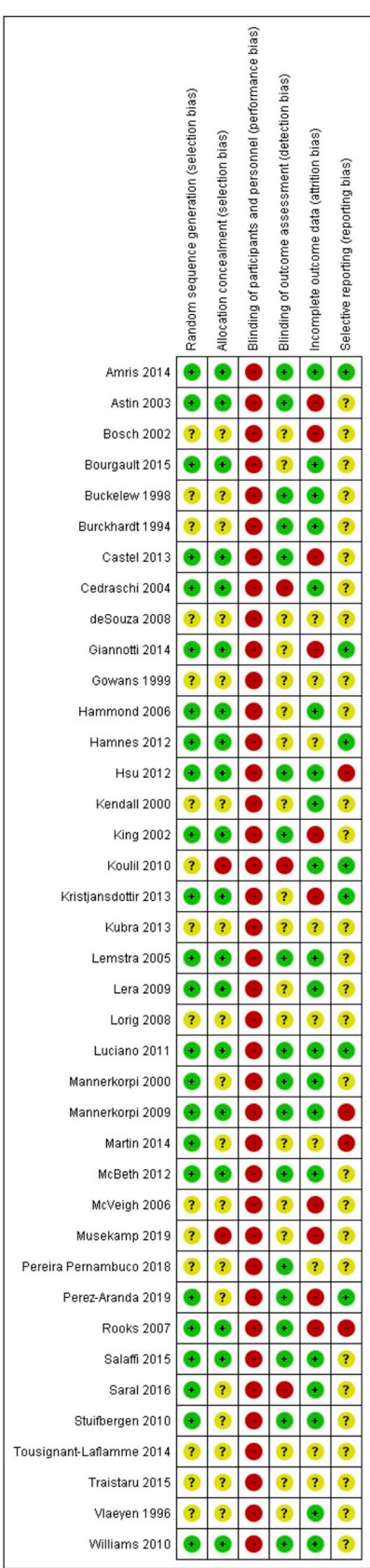

**Fig 3. Risk of bias scoring for all included completed studies.**

and were rated as unclear. Three trials were rated as high risk, noting that it was not possible to blind assessors [48, 64, 67].

**Incomplete outcome data.**    Nineteen studies were rated as low risk, reporting follow-up of higher than 75% and following an ITT principle [27] Ten studies were rated as unclear risk with follow-up above 75% but not enough details to determine if ITT had been followed. Ten studies were rated as high risk, with follow-up bellow 75% and (or) only a per-protocol analysis reported [32, 37, 42, 51, 53, 60, 62, 68–70].

**Selective reporting.**    Seven studies were rated as low risk, with published trial registration entries aligning with reported outcomes. Twenty-eight studies were rated as unclear, with no registry entry available or retrospective entries. Four studies were rated as high risk, with differing outcomes reported in the final report compared to the registry entry [35, 55, 65, 68].

## GRADE study quality rating

The quality of the evidence for the review outcomes was rated as low in most cases. Tables 2 and 3 show our summary of findings for our two main comparisons (self-management vs. usual care and self-management vs. active comparison). Most outcomes were rated down with a combination of serious limitations (risk of bias) and either serious inconsistency, or serious imprecision. Further details can be found in the full evidence profiles provided as S4 and S5 Files.

## Effects of multicomponent self-management intervention vs. waitlist/no treatment/usual care

**Primary outcome: Physical function.**    Fourteen studies reported objective and/or subjective physical function outcomes. Objective outcomes are described first. Five studies reported objective outcomes. Due to differences in data presentation, these objective outcomes were not meta-analysed: Four studies used the 6-minute walk test (6MWT) as an objective measure. One study reported descriptive data showing small increased distance walked for the intervention group and reduced distance for the wait list control group at 12-week follow-up, but did not report between-group significance tests [58]. Two studies reported significantly increased walking distances compared to controls in the short-term [49, 62]. One study reported within group improvements in walking distance in the intervention group, and a lack of within group change in the control group [70]. Regarding long-term outcomes, one study reported maintenance of increased walking distances at six-month follow-up [70]. One study used objective measures of motor ability, moving and adapting actions, [31]. The authors reported significant benefit in these measures at six-month follow-up for the intervention group compared to waitlist controls.

Studies that presented subjective physical function outcomes used a range of measures. The most commonly used included the Fibromyalgia Impact Questionnaire (FIQ) physical functioning subscale [48, 49, 52, 54, 58, 71], and the physical functioning item from the SF-36 [46, 68]. Some studies used the SF-36/8 physical component summary as a primary measure of physical function [32, 55]. Consequently, for consistently, where studies did not present alternative physical functioning measures, but did present an SF-36/12/8 physical component summary score this was used as a measure of physical functioning for those studies.

Of the studies that presented subjective physical function outcomes, six studies did not have data sufficient for meta-analysis of these outcomes. Results from these six studies varied: Three studies reported no significant differences between intervention groups and controls in the short or long-term [31, 38, 49]. Two studies reported greater within group change in the short-term for intervention groups than controls but did not directly compare with

**Table 2. Summary of findings: Self-management interventions for chronic widespread pain (CWP) including fibromyalgia.**

| Self-management interventions compared with usual care or no treatment controls | | | |
|---|---|---|---|
| **Patient or population: Adults with CWP or fibromyalgia** | | | |
| **Setting: outpatient or inpatient** | | | |
| **Intervention: Self-management intervention** | | | |
| **Comparison: Usual care or no treatment controls** | | | |
| **Outcomes** | **Effects (SMD/Narrative)** | **No. of Participants (studies)** | **Quality of the evidence (GRADE)** |
| Objective physical function–short-term | One study reports significantly improved function in SM intervention participants compared to controls. One study reports significant improvements in function in SM intervention group compared to controls in complete case subgroup analysis only. Two studies report some limited evidence of within group improvements in function in SM intervention group contrasted with little within group change in the control. | 221 (4) | ⊕ Very low[a] |
| Objective physical function–long-term | One study reports significant improvements in function in the SM intervention group compared to control, one study shows significant within group improvements in the SM intervention group contrasted to a lack of within group change in the control group. | 206 (2) | ⊕⊕ Low[b] |
| Self-reported physical function–short-term | SMD: 0.42 (0.20, 0.64). 5 RCTs | 723 (9) | ⊕⊕ Low[b] |
| | One study reported significant improvements in subjective function in SM intervention compared to controls. Two studies report within group improvements in the SM intervention group only. One study reports no significant differences between SM intervention and control group. 4 RCTs | | |
| Self-reported physical function–long-term | SMD: 0.36 (0.20, 0.53). 8 RCTs | 990 (10) | ⊕⊕ Low[b] |
| | Both studies showed no significant difference between SM intervention and control. 2 RCTs. | | |
| Pain–short-term | SMD: -0.49 (-0.70, -0.27). 6 RCTs. | 1049 (12) | ⊕⊕ Low[c] |
| | Three studies reported significant reductions in pain in SM intervention compared to controls. Three studies reported no significant differences between SM interventions and controls. 6 RCTs. | | |
| Pain–long-term | SMD: -0.38 (CI -0.58, -0.19). 9 RCTs. | 1135 (12) | ⊕⊕ Low[c] |
| | Three studies showed no significant difference between SM intervention and control. | | |

*Abbreviations*: SMD, standardised mean difference; SM, Self-management; RCT, randomised controlled trial; GRADE, Grading of Recommendations Assessment, Development, and Evaluation

[a] Rated down for limitations, inconsistency and imprecision.

[b] Rated down for limitations and imprecision.

[c] Rated down for limitations and inconsistency

significance tests [58, 71]. One study reported significant improvement in subjective physical function at six-week follow-up compared to a usual care control group [59].

Eleven studies included data sufficient for meta-analysis of subjective physical outcome measures. Analysis showed a moderate improvement in physical function in favour of the interventions for both short-term outcomes (n = 473, SMD 0.42, 95% CI 0.20, 0.64; P = 0.0002) and long-term outcomes (n = 724, SMD 0.36, 95% CI 0.20, 0.53; P < 0.0001). There were low levels of heterogeneity in the comparisons for both the short-term ($I^2$ = 22%) and long-term analysis ($I^2$ = 14%). See Fig 4.

**Primary outcome: Pain.** Twenty one studies reported measures of pain severity with short-term or long-term follow-up. The most common measure used was a pain visual analogue scale. Ten studies did not provide data sufficient to be included in a meta-analysis: Three studies [47, 49, 63] reported no significant differences in pain outcome between intervention and control conditions at short-term follow-up. Four studies reported significant reductions in pain severity compared to control groups at short-term follow-up [39, 43, 59, 69]. Amris et al., [31] and Lorig et al., [31, 38] measured pain with a focus on long-term outcomes; they

**Table 3. Summary of findings: Self-management interventions for chronic widespread pain (CWP) including fibromyalgia.**

| Self-management interventions compared with active comparisons | | | |
| --- | --- | --- | --- |
| **Patient or population: Adults with CWP or fibromyalgia** | | | |
| **Setting: Outpatient or inpatient** | | | |
| **Intervention: Self-management intervention** | | | |
| **Comparison: Active comparisons** | | | |
| **Outcomes** | **Effects (SMD/Narrative)** | **No. of Participants (studies)** | **Quality of the evidence (GRADE)** |
| Objective physical function–short term | All four studies reported no significant differences between SM intervention and active comparison. | 481 (4) | ⊕⊕ Low[a] |
| Objective physical function–long term | One study reported no significant improvements in function in the SM intervention group compared to an active comparison. One study reported significant within group improvement in the intervention condition alone. | 249 (2) | ⊕ Very low[b] |
| Self-reported physical function–short term | SMD: 0.12 (-0.06, 0.30). 5 RCTs. | 665 (7) | ⊕⊕ Low[a] |
| | One study reported significant improvement function within SM intervention group, and not within the active comparison group, but they were not directly compared. One study reported the SM intervention group and an exercise group showed improvements compared to an active control. 2 RCTs. | | |
| Self-reported physical function–long term | SMD: -0.01 (-0.17, 0.16). 6 RCTs | 1357(10) | ⊕⊕ Low[a] |
| | Three studies showed no difference between SM intervention and active comparison. One study showed within group improvements in function in the SM intervention and no within group improvements in the active comparison condition. 4 RCTs. | | |
| Pain–short term | SMD: 0.04 (-0.28, 0.21). 3 RCTs | 510(5) | ⊕⊕ Low[a] |
| | One study found that the SM intervention reduced pain compared to an active comparison, one study reported no difference between the SM intervention and active comparison. 2 RCTs. | | |
| Pain–long term | SMD: 0.10 (CI -0.41, 0.34). 5 RCTs. | 1011 (8) | ⊕⊕ Low[a] |
| | One study showed no significant difference between SM intervention and active control. One study showed no within group difference in pain and did not compared groups. One RCT did not directly compare active comparisons with SM intervention, but reported neither the SM intervention or the active comparisons were more effective than usual care in reducing pain in the long term. 3 RCTs. | | |

*Abbreviations*: SMD, standardised mean difference; SM, Self-management; RCT, randomised controlled trial; GRADE, Grading of Recommendations Assessment, Development, and Evaluation

[a] Rated down for limitations and imprecision

[b] Rated down for limitations, imprecision and inconsistency.

reported no significant differences between interventions and controls on pain at six-month [31, 38] and 12-month follow-up [38]. One study used a chronic pain grade score and reported no difference between intervention and control condition at 9 months [33].

Eleven studies provided data sufficient for meta-analysis on pain outcomes. In the short-term, the analysis participants showed a moderate reduction in pain in favour of the intervention compared to the control (n = 628, SMD -0.49, 95% CI -0.70, -0.27; P <0.00001). In the long-term, the analysis also showed a moderate reduction in pain in favour of the intervention (n = 790, SMD -0.38, 95% CI -0.58, -0.19; P = 0.00001). There was moderate heterogeneity in the comparisons in the short-term ($I^2$ = 36%) and long-term analysis ($I^2$ = 40%). See Fig 5.

**Disease specific measure–Fibromyalgia Impact Questionnaire FIQ.** The most commonly included disease specific measure was a total score on Fibromyalgia Impact Questionnaire (FIQ, higher scores indicate greater negative impact). The FIQ total measure was included in 20 studies. Seven studies did not include sufficient data for meta-analyses: Three studies reported significant reductions in total FIQ score compared to control conditions at short-term follow-up [43, 56, 71]. Three studies reported no significant differences between

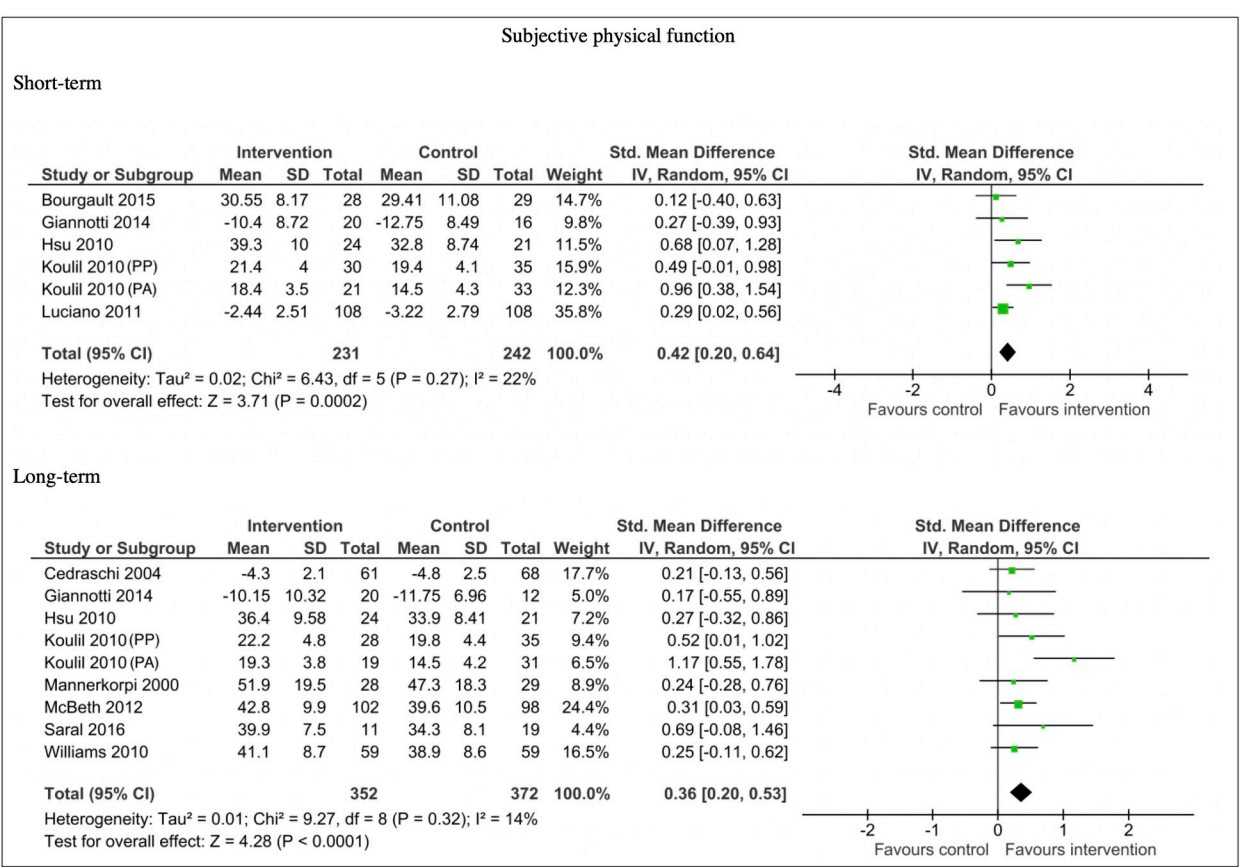

**Fig 4. Forest plots of comparisons for subjective physical function for intervention compared to waitlist/no treatment/usual care controls.**

intervention and control conditions at short-term follow-up [40, 44, 58]. Amris et al., [31] reported no significant difference between the intervention and control condition at six-month follow-up.

Thirteen studies provided FIQ total data that could be pooled for meta-analysis. The analysis showed a moderate reduction in overall impact of fibromyalgia as measured by the total FIQ score in favour of the intervention in the short-term (n = 853, MD -6.64, 95% CI -11.45, -1.83; P = 0.007). For this short-term analysis mean difference was used, as the same scaling was used in all studies. For comparability, the short-term mean difference is equivalent to a standardised effect size of -0.39 (-0.68–0.10). A moderate reduction in overall impact was seen in the long-term analysis (n = 736, SMD -0.49, 95% CI -0.64, -0.34; P <0.00001). There was substantial heterogeneity in the short-term comparisons ($I^2$ = 75%), however, there was little heterogeneity in the long-term comparisons ($I^2$ = 0%). See Fig 6.

**Fatigue.** Thirteen studies measured fatigue. The data from four studies could not be pooled for meta-analysis: Gowens et al., [49] reported significant reductions in fatigue in the morning [FIQ subscale] in the intervention condition, compared to the control at post-intervention follow-up. However, they found no significant differences in general fatigue between the intervention and control condition at same time point. Salaffi et al., [71] reported significantly reduced fatigue in the intervention compared with control at post-treatment follow-up, using a mean of time integrated values calculated for each patient. Two studies found no significant differences between the intervention and control conditions at 12-week follow-up [58] and six-month follow-up [38].

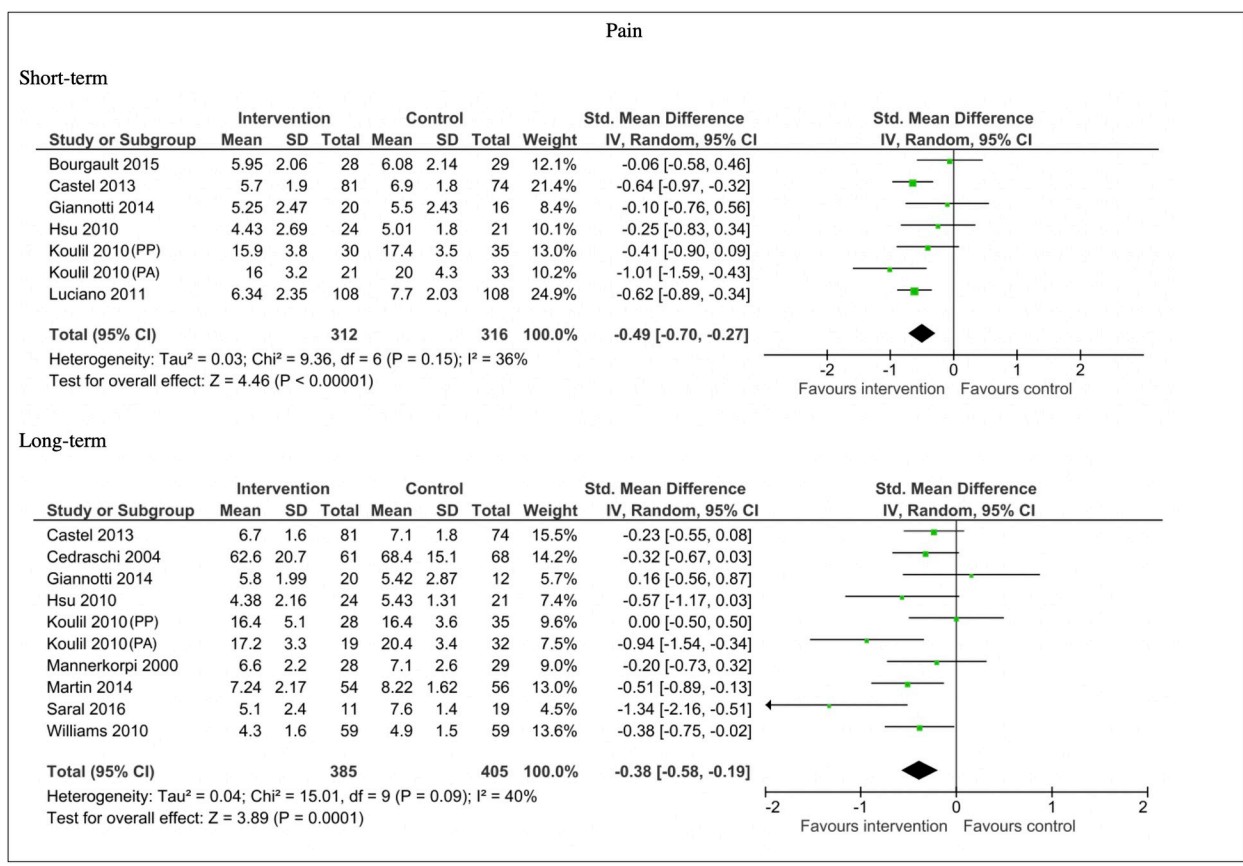

**Fig 5. Forest plots of comparisons for pain measures for intervention compared to waitlist/no treatment/usual care controls.**

Nine studies included data sufficient for meta-analysis. Both the short-term and long-term analysis showed a reduction in fatigue in favour of the intervention. This effect was moderate in the short term (n = 416, SMD -0.75, 95% CI -1.17, -0.33; P = 0.0004), and somewhat smaller in the longer term (n = 715, SMD -0.35, 95% CI -0.55, -0.16; P = 0.0003). Substantial heterogeneity was present in the short-term analysis ($I^2$ = 71%), heterogeneity was moderate for the long-term analysis ($I^2$ = 33%). See Fig 7.

**Global health measures.** Six studies included measures of global improvement. Data included for these studies was not sufficient for meta-analysis. Bourgault et al., [47] reported that the likelihood of reporting overall improvement in pain, level of functioning and quality of life, was higher in the intervention condition, compared to the controls at post-treatment follow-up. Tousignant-Laflamme et al., [40] reported greater overall improvement on global impression of change ratings of pain, function and quality of life in the intervention group, compared to the control group at three months after delivery of the intervention. Williams et al., [46] used a single item global impression of change measure and reported that 57% of those in the intervention group completing this measure reported improvements at 6-month follow-up, compared to 21% of those in the control group. McBeth et al., [33] also used a single item measure of global health since entering the trial. They reported that the percentage reporting a positive outcome at 9 months was 8% in the treatment as usual group, compared to 37% in the intervention group. Lorig et al., [38], used a measure of self-reported global health and reported no significant differences between an online self-management intervention and a usual care control at one year.

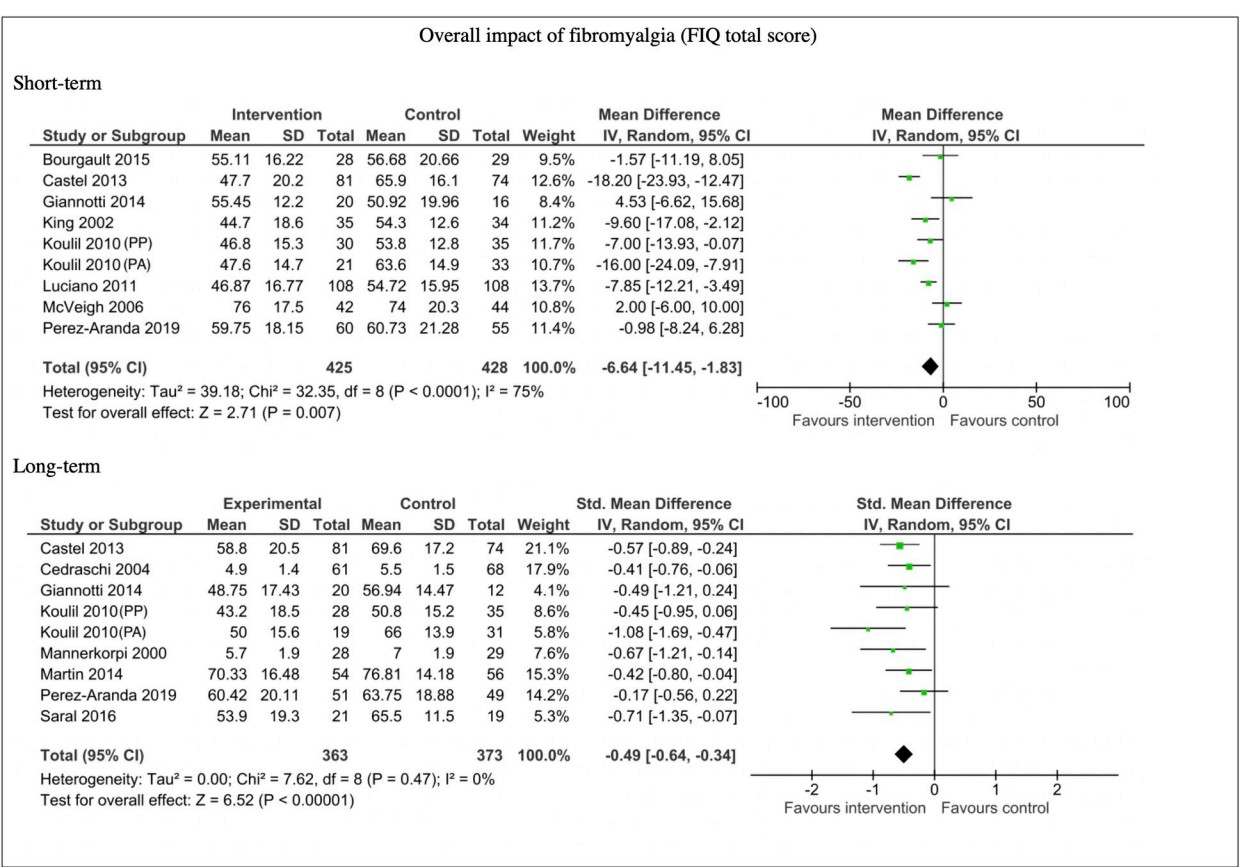

**Fig 6. Forest plots of comparisons for impact measures (total FIQ score) for intervention compared to waitlist/no treatment/usual care controls.**

**Quality of life.** Nine studies reported quality of life measures. Seven reported composite measures with subscales that aligned with other outcomes (e.g. physical functioning, mental functioning, pain) and consequently were analysed with these outcomes. Two studies used total scores on quality of life scales. Burckhardt et al., [58] measured quality of life with the Quality of Life Scale [72]. They reported significant differences in the intervention group at post-treatment follow-up due to a drop in QoL in the control group. Traistaru et al., [43] measured quality of life with a scale developed by Spitzer et al., [73]. They reported significant within group improvements in quality of life in the intervention group at post-treatment follow-up.

**Mental health.** Sixteen studies included mental health measures. Eleven studied included measures of depression or mood, three studies measured distress more broadly using the General Health Questionnaire 6 (GHQ-6) [44], or used a composite score of the Hospital Anxiety and Depression Scale (HADS) anxiety and depression subscales [53, 60]. One study used the mental health component of the Short Form-36 scale (SF-36) [55], and one study used the Psychological General Well-Being index [48].

Five studies did not contain data sufficient for meta-analysis: Lemstra et al., [59] reported significant reductions in depression compared to the control group at six-week follow-up. Three studies reported no significant differences at short term follow-up points in measures of depression [49, 58] and a measure of distress [44] between intervention and control conditions. Amris et al., [31] measured depression at six months and reported no significant differences between the intervention and the control.

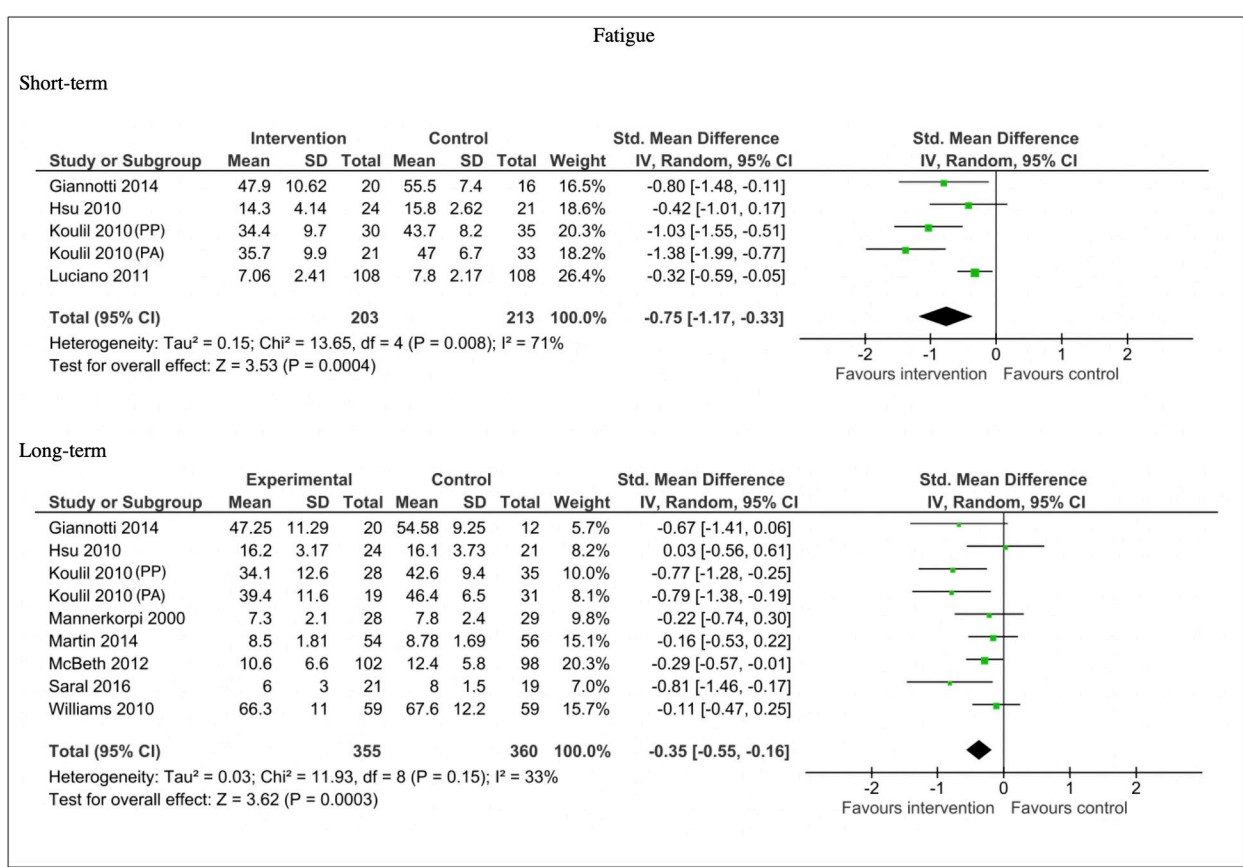

**Fig 7. Forest plots of comparisons for fatigue measures for intervention compared to waitlist/no treatment/usual care controls.**

Twelve studies included mental health data that could be pooled for meta-analysis.

The results showed improvements in mental health, primarily through reductions in depression and distress, favouring the intervention group compared the controls in the short term (n = 707, SMD -0.40, 95% CI -0.66, -0.14; P = 0.003), and in the long-term (n = 1069, SMD -0.29, 95% CI -0.51, -0.08; P = 0.007). Both short-term and long-term comparisons showed substantial heterogeneity, $I^2$ = 63% and $I^2$ = 65% respectively. See Fig 8. The long-term mental health analysis had 10 unique studies, thus possible publication bias was explored by assessing funnel plot asymmetry (see Fig 9). Egger's test for small study effects was not significant (p = 0.10) again suggesting no evidence of funnel plot asymmetry.

**Harms.** Six studies explicitly noted the presence or absence of harms or negative effects of the intervention. Three studies reported an absence of harms or related adverse events/reactions [33, 70, 71]. Three studies reported negative effects of the interventions. Perez-Aranda et al., [60] described that in the multicomponent self-management group one participant reported severe tension and slight headaches. Seven other participants reported experiencing symptoms including tension, fatigue and headache, but these were reported as infrequent, transient and/or low intensity. Saral et al., [67] reported occasional, mild increases in pain after some exercise sessions. Lemstra et al., [59] stated that 20 people in the intervention group reported minor musculoskeletal pain as a side effect.

**Healthcare utilisation and medication usage.** Two studies included direct comparisons of health care utilisation or medication use between a multicomponent self-management group and a waitlist/usual care control. Luciano et al. [54] reported reduced health care costs

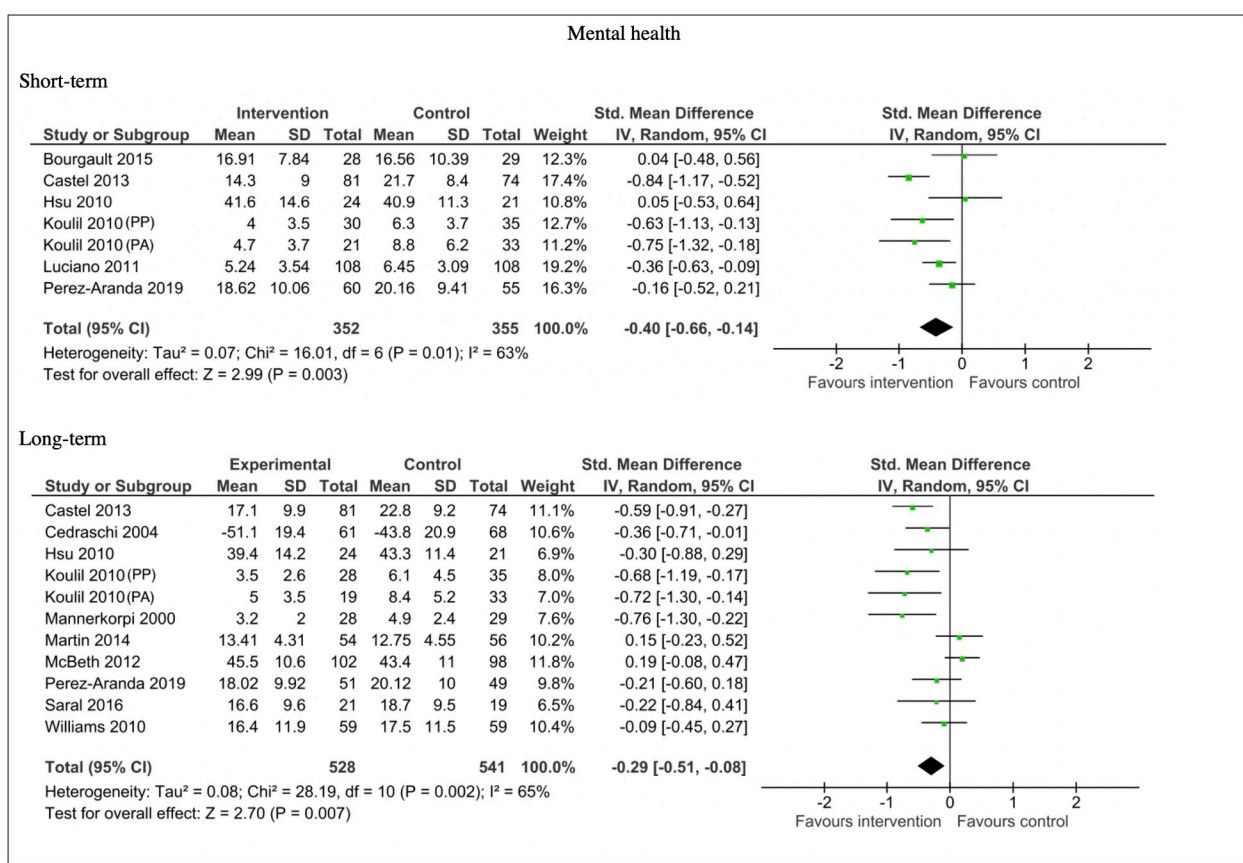

**Fig 8. Forest plots of comparisons for mental health measures for intervention compared to waitlist/no treatment/usual care controls.**

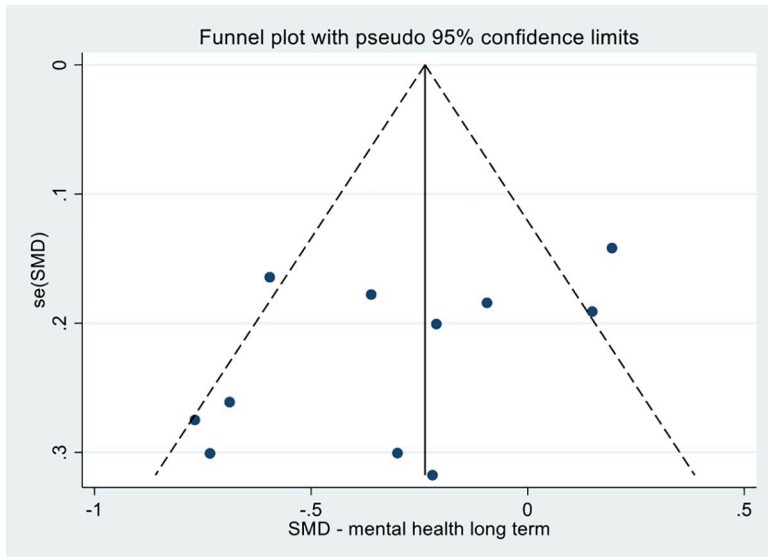

**Fig 9. Funnel plot of studies including mental health as a long-term outcome.**

in the intervention condition compared to usual care. They concluded that there was between 98% and 95% probability that supplementing usual care with their intervention was more cost effective than usual care alone. Lemstra et al., [59] reported no significant difference in medication prescription and non-prescription drug use in the last 30 days between the intervention and control group at post-treatment follow-up.

### Effect of self-management intervention vs. active comparisons

**Physical function.**  Eleven studies had comparisons of self-management interventions vs. additional active interventions and also included physical function outcomes. Seven studies did not include data sufficient for meta-analysis:

Astin et al., [51] compared mindfulness training combined with Qigong exercises to an educational support intervention. They found that both intervention and active control participants increased their 6-minute walk distance at 8 and 24 weeks; there were no significant differences between groups. Burckhardt et al., [58] compared a self-management educational programme combined with physical exercise to the self-management education programme alone. They reported improvements in both groups' 6-minute walk distance at post-treatment follow-up; there were no significant differences between groups. Burckhardt et al., [58] also included a subjective measure of function. The reported significant within group improvements in the combined group, but no within group differences in education alone at post treatment. They were not directly compared. Buckelew et al., [57] conducted a four-arm trial comparing biofeedback relaxation combined with exercise, to a biofeedback alone group, an exercise alone group, and an attention control group. They reported that the combined intervention group had significant improvements in physical activity (measured by the Arthritis Impact Measurement Scales (AIMS), physical activity subscale) compared to the attention control group at post-treatment but this difference was not significant at two-year follow-up. There were no within-group improvements in physical activity in the biofeedback alone and attention control group at any time points. King et al., [62] compared a combined education and exercise intervention, with exercise alone and education alone. They reported that both the combined intervention and the exercise alone, but not the education alone group, significantly increased their 6-minute walk test difference at short-term post-treatment follow-up. Mannerkorpi et al., [35] compared a combined education and pool exercise intervention with education alone. They reported that the combined intervention led to increases in 6-minute walk test distance compared to the education alone control at post-treatment follow-up, however this difference was not statistically significant (p = 0.067). At 12-month follow-up they report significant within group improvement for the intervention group, but no within group improvement for the education alone group. McBeth et al., [33] included combined telephone CBT and exercise arm, a telephone CBT alone arm, and exercise alone arm in their RCT. They reported that the combined intervention and exercise intervention led to significant improvements in SF-36 physical component score at 9 months compared to treatment as usual, whereas this was not case for telephone CBT alone. Hammond et al., [52] compared a patient education intervention that included physical exercise to a relaxation group. They reported little change in physical function (measured using the physical function scale of the FIQ), in both groups. There were no statistical differences between groups at 8-month follow-up.

Six studies included data that could be pooled for meta-analysis (see S3 File and Table 1 for details of interventions and comparators). The analysis showed that there was no significant difference in physical function outcomes between multicomponent self-management interventions and active comparators at short-term (n = 490, SMD 0.12, 95% CI -0.06, -0.30; P = 0.20) and long-term follow-up points (n = 451, SMD -0.10, 95% CI -0.33, 0.13; P <0.80).

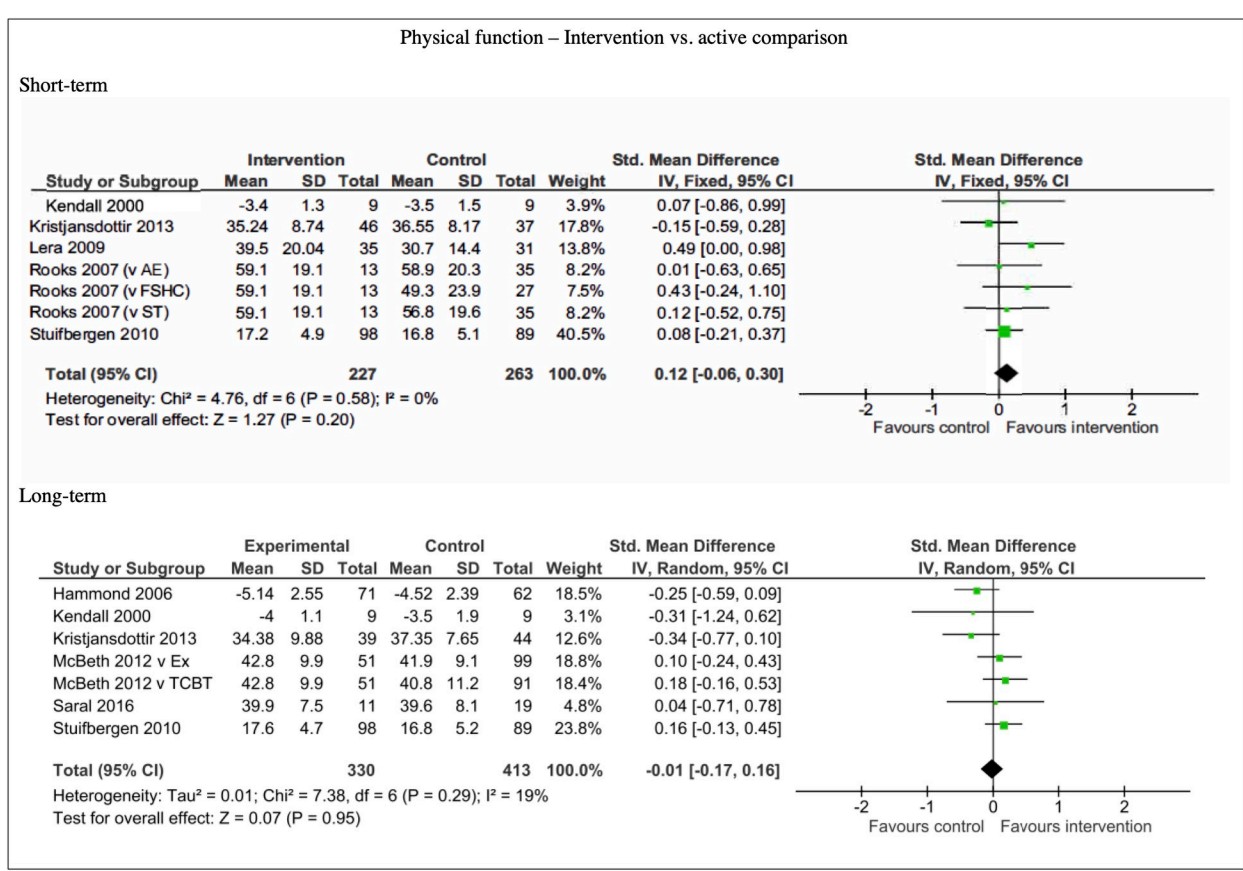

**Fig 10. Forest plots of comparisons for self-reported physical function measures for intervention compared to active controls.**

Short-term comparisons showed little heterogeneity ($I^2 = 0\%$), whereas long-term comparisons showed moderate heterogeneity ($I^2 = 24\%$). See Fig 10.

**Pain.** Ten studies with an active comparison groups included a pain measure. Four studies did not contain sufficient data for meta-analysis: Mannerkorpi et al., [35] reported significant differences in pain between a combined education and pool exercise intervention and education alone at post-treatment follow-up. At 12 months, they reported no within group significant differences, and did not actively compare groups. Burckhardt et al., [58] found no significant differences in pain at post-treatment follow-up between their self-management educational programme plus exercise, and the self-management educational programme alone. Hammond et al., [52] reported no significant differences in pain between a patient education intervention that included physical exercise and a relaxation group at 8-month follow-up. McBeth et al., [33] did not directly compare the active arms in their trial, however they did report that all 3 active arms; telephone CBT plus exercise, telephone CBT alone, and exercise alone, did not differ significantly from usual care in reducing chronic pain grade ratings at 9-month. In their four-armed trial, Buckelew et al. reported no between group significant differences in pain at post-treatment or two-year follow-up when comparing combined biofeedback with exercise, biofeedback alone, exercise alone and active control, at post-treatment and two-year follow-up.

Five studies included data on pain outcomes that could be pooled for meta-analysis. The analysis showed no significant differences between the multicomponent self-management interventions and the active comparators at short-term follow-up (n = 288, SMD -0.04, 95%

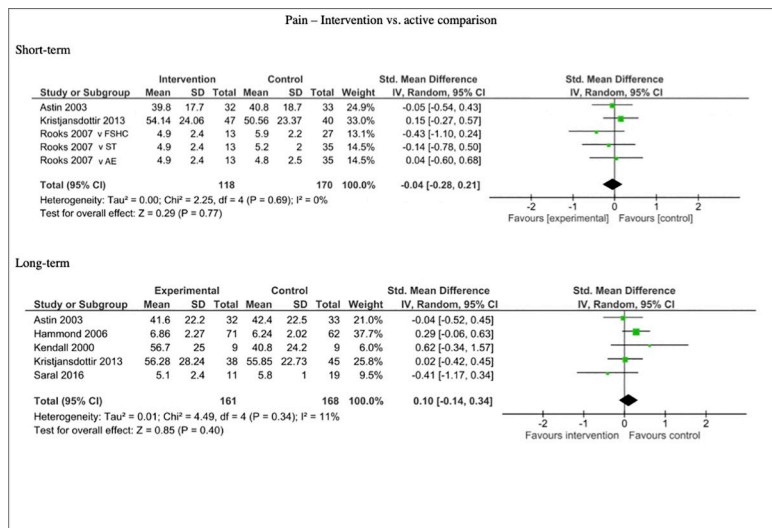

**Fig 11. Forest plots of comparisons for self-reported pain measures for intervention compared to active controls.**

CI -0.28, 0.21; P = 0.77) and long-term follow-up (n = 329, SMD 0.10, 95% CI -0.14, 0.34; P = 0.40). Short-term comparisons showed low heterogeneity ($I^2$ = 0%), as did long-term comparisons ($I^2$ = 11%). See Fig 11.

**Protocols and registry entries.** Four protocol papers were included describing ongoing studies dated between 2018–2020. All four used fibromyalgia as a classification; three studies used the American College of Rheumatology criteria (2010–11) and one study used the ICD criteria (M79.7). The components comprising the primary intervention were as follows: Three of four studies included physical activity [74–76] which ranged from aerobic, strengthening and stretching activities, to Nordic walking and yoga. Three of four studies included a psychological component. Techniques included discovery through intentionally attending to emotional cognitive and bodily experiences [77], and cognitive behavioural strategies [74–76]. Three of four studies included a medical education component; ranging from neuroscience education [74, 76], pharmacology of pain [75]. Two of four studies included mind-body techniques, primarily mindfulness-based approaches [76, 77]. One study included a lifestyle component, covering management of sleep problems and nutrition [75]. All four studies compared the central interventions to usual care. Group delivery was used in two studies [75, 76], and group plus individual delivery was used in the remaining two studies [74, 77]. Follow-up ranged from 12 weeks [76] to 15 months [75].

Eleven unique trial registry entries were identified that met the inclusion criteria. Of those, six were listed as completed without results posted: Dates ranged from 1999 to 2018, and all six used fibromyalgia as their diagnostic classification [78–83]. Physical activity was a component in the self-management intervention in five of six of these studies [78–80, 82, 83]. Medical education featured in three of six of the interventions [81–83], with a psychological component also reported in three of six interventions [78–80]. One of these interventions featured a lifestyle component [81]. In one case delivery was not clear [79], in five of six studies, groups were used to deliver the interventions. All interventions in these studies were compared to one or more active comparators, and follow-up ranged from one month to 12 months.

Five registry entries were listed as ongoing, with dates ranging from 2016 to 2020: All used fibromyalgia as their diagnostic classification. Physical activity was a component of all five interventions [79, 84–88]. Medical education was a part of four of five interventions [82, 84,

86, 87]. Two of five interventions featured a mind-body component [84, 85], and one of five featured a psychological component [87]. Only one of the ongoing studies described a lifestyle advice component as part of their intervention [84]. Three of five studies compared the intervention to a usual care control [84, 85, 87], two studies compared interventions to active comparators [84, 85, 87]. Regarding delivery, three studies reported group-based delivery [82, 84, 86]; one study reported using a combination of group and individual delivery [85]; and one used digital delivery [via mobile phones] with addition support from a health professional coach [87]. Follow-up periods ranged from three to 12-months.

## Discussion

We aimed to determine the effectiveness of interventions applying self-management principles for individuals with CWP including fibromyalgia. Despite some variability, self-management interventions improved self-reported and objective physical function in the short and long-term compared to waitlist or usual care controls. With regard to pain outcomes, there was greater variability in the studies narratively reviewed, however, the meta-analytic results showed self-management interventions produced a moderate reduction in pain in both the short and long-term. The fibromyalgia impact questionnaire (FIQ) was the most consistently used secondary outcome measure across studies. Findings for the FIQ were similar to pain; with variability in the findings of the six narratively reviewed studies, and the meta-analysis of 13 studies showing moderate improvements for self-management over waitlist / usual care controls in the short and long-term. The remaining secondary outcomes investigated, including fatigue, mental health and quality of life, shared a similar pattern; providing some indication of benefit for the self-management intervention compared to controls, in the midst of considerable variability. When self-management interventions were compared to active interventions on physical function and pain, the majority of studies reported no significant differences between groups.

These findings pertaining to effectiveness need to be considered in the context of study quality: Risk of bias was unclear across multiple domains for the majority of included studies. Only two studies were rated as low risk on all domains apart from blinding of participants (not possible in behavioural trials) [31, 54]. Additionally, our GRADE rating for our primary outcomes was low in most cases, reducing certainty. Nevertheless, there were broadly consistent effects in favour of the interventions in the studies included in the meta-analyses across a range of outcomes. This indicates that interventions applying self-management principles can be effective in both the short and long-term for CWP including fibromyalgia when compared to waitlist or usual care controls.

The present review has some notable differences from Häuser et al.'s 2009 [19] review of multicomponent treatments: Our focus on self-management and use of an aligning definition [21], meant that we excluded studies where core intervention components were 'passive', such as patients receiving treatments rather than learning skills that could be applied beyond the intervention e.g. massage and spa treatments. Additionally, using Miles et al.'s [21] definition we included studies of interventions including at least two of any five components [psychological, lifestyle, physical activity, medical education, and mind-body therapies]. Häuser et al. focused exclusively on psychological or educational approaches, plus an exercise component. Regarding effectiveness, our findings are similar to Häuser et al.'s [19]; broadly showing that multicomponent self-management interventions can be effective in improving physical function in the short- and long-term, as well as reducing pain in the short-term for those experiencing CWP including fibromyalgia. Importantly, we did find some evidence of long-term effects of self-management interventions in reducing pain in our meta-analysis and on

the wider impact of the condition (measured via FIQ), diverging from Häuser et al.'s findings of reduced long-term effects. Our findings also shared a similar pattern to Cochrane reviews of CBT [17] and exercise [16] for fibromyalgia. Bernardy et al. [17] concluded that CBT produces small effects on pain and mood in the short and long-term when compared to no treatment/ usual care controls. Bidonde et al.'s [16] review suggested that exercise produces improved function in the short and long-term and reduced pain in the short term when compared to no treatment/usual care controls. However, similar to our review, both Bidonde et al. [16] and Bernardy et al. [17] found no differences when CBT and exercise were compared to active interventions.

Finding few differences in outcomes between different interventions for chronic pain is common. It is reported across a range of interventions [16, 17, 89, 90], and pain types including back pain [89, 91], neck pain [92] and mixed chronic pain [93]. Whilst our review echoes these findings on effectiveness, self-management may have important advantages regarding applicability. In self-management interventions, patients learn about a range of approaches e.g. physical activity, mind-body techniques and psychological strategies, potentially accommodating a broader range of preferences and increasing opportunities for engagement [94].

Primary care is increasingly recommended as the most appropriate medical setting for managing CWP [6, 95, 96]. Self-management interventions thus need to be initiated from, and integrated with, primary care provision. Our review of ongoing study protocols and registry records demonstrates there is considerable work continuing on general self-management interventions. However, intervention content and formats described are very similar to the completed studies that we have reviewed, albeit with welcome increases in quality and rigour. Going forward, there is a need to develop novel self-management interventions that can be specifically integrated into primary care and made widely accessible. This need for scalable self-management is likely to increase with the COVID-19 pandemic. Clauw et al. [97] and Kemp et al. [98] highlight multiple biopsychosocial routes through which the pandemic is likely to increase presentation of chronic pain, including CWP. Patients with COVID-related widespread pain are likely to consult in primary care [99], and Kemp et al. [98] directly call for innovation and development of accessible self-management programmes.

Future work on development of self-management interventions for CWP should heed recent advice on development of psychological interventions for chronic pain, calling for a step change to avoid research waste [100]. The majority of studies we reviewed did not explicitly reference theory as underpinning interventions. Use of theory based on existing models of behaviour and pain maintenance to support selection of intervention content, coupled with more overt theorising post-trial based on results, should help to increase understanding of the common variability reported in trial outcomes. Theorising should go beyond treatment specific models and include and account for overarching common processes that appear consistently important in CWP [101]. Understanding how these common processes can best be explicated and capitalised on should increase intervention effectiveness [102]. Complexity should also be built into this modelling, moving beyond linear theory [103, 104] and working to account for how context may impact on individual variation [105]. Consideration and planning for individual variation in response and preferences is likely to be particularly important in CWP [106]. Alongside theorising and necessary evidence synthesis, there is a need for a rich understanding of patient experience [of both illness and management] [107]. This will ensure new interventions are developed where potentially effective components are delivered in a persuasive, engaging and accessible format [94, 107, 108].

There are some limitations to consider with our review. The definition of self-management we used led to the inclusion of broad range of interventions, which may have increased variability across outcomes. Nevertheless, all intervention included core self-management

principles, and our aim was to review this field and describe the range of approaches used. The majority of studies had unclear risk of bias and due to this limited variability, we did not conduct a sensitivity analysis based on quality. Future research must focus on ensuring methodological rigour and detailed reporting. There were not enough studies that used alternatives to group-based delivery of the intervention to determine the impact of delivery modality on effectiveness. Further work will be required on alternatives such as internet, app or telephone-based delivery to draw conclusions in this area. We found a range of diagnostic criteria used for CWP and fibromyalgia, which may have increased variability in the clinical characteristics of those in the studies review. Relatedly, the majority of studies used fibromyalgia specifically as a diagnostic category, rather than the broader CWP. As such, these finding are primarily applicable to those for whom CWP is a symptom of fibromyalgia syndrome. Further research is needed to determine how results may vary if CWP was primarily used as diagnostic entry criteria.

## Conclusions

Research reviewed suggests self-management interventions for CWP including fibromyalgia can be effective at improving physical function and reducing pain in the short and long-term. However, in our review study quality was often limited by unclear risk of bias, and the quality of evidence by outcome was low. Future research should focus on increasing methodological quality and on developing accessible self-management interventions based on evidence, theory and patient experience.

## Supporting information

**S1 File. Protocol for systematic review.**
(PDF)

**S2 File. MEDLINE search strategy.**
(PDF)

**S3 File. Study characteristics table.**
(PDF)

**S4 File. Evidence profile table for self-management vs. usual care/no treatment controls.**
(PDF)

**S5 File. Evidence profile table for self-management vs. active comparisons.**
(PDF)

**S6 File. PRISMA checklist for CWP systematic review.**
(PDF)

## Author Contributions

**Conceptualization:** Adam W. A. Geraghty, Emma Maund, Miriam Santer, Hazel Everitt, Cathy Price, Tamar Pincus, Michael Moore, Paul Little, Beth Stuart.

**Data curation:** Adam W. A. Geraghty, Emma Maund, David Newell, Rachel West, Beth Stuart.

**Formal analysis:** Adam W. A. Geraghty, Beth Stuart.

**Funding acquisition:** Adam W. A. Geraghty, Miriam Santer, Hazel Everitt, Beth Stuart.

**Investigation:** Adam W. A. Geraghty, Emma Maund.

**Methodology:** Adam W. A. Geraghty, Emma Maund, David Newell, Cathy Price, Tamar Pincus, Michael Moore, Paul Little, Rachel West, Beth Stuart.

**Project administration:** Adam W. A. Geraghty.

**Resources:** Adam W. A. Geraghty.

**Visualization:** Beth Stuart.

**Writing – original draft:** Adam W. A. Geraghty, Emma Maund, David Newell, Miriam Santer, Hazel Everitt, Cathy Price, Tamar Pincus, Michael Moore, Paul Little, Rachel West, Beth Stuart.

**Writing – review & editing:** Adam W. A. Geraghty, Emma Maund, David Newell, Miriam Santer, Hazel Everitt, Cathy Price, Tamar Pincus, Michael Moore, Paul Little, Rachel West, Beth Stuart.

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
