## [Editor Report · Decision Letter 0]

17 Nov 2020

PONE-D-20-35048

Self-management for chronic widespread pain including fibromyalgia: A systematic review and meta-analysis

PLOS ONE

Dear Dr. Geraghty,

Thank you for submitting your manuscript to PLOS ONE. After careful consideration, we feel that it has merit but does not fully meet PLOS ONE’s publication criteria as it currently stands. Therefore, we invite you to submit a revised version of the manuscript that addresses the points raised during the review process.

see below.

We look forward to receiving your revised manuscript.

Kind regards,

Tim Mathes

Academic Editor

PLOS ONE

Journal Requirements:

2. Please provide a quality a publication bias assessment on the studies included in the systematic review.

Additional Editor Comments:

Your wrote that you use a Random-Effects meta-analysis. However, the figures show that often the fixed-effects model was used. Probably only forgotten to change in RevMan. Before we can send the article to peer review this should be revised.
---

## [Author Response · Author response to Decision Letter 0]

6 Jan 2021

Additional Editor Comments:

"Your wrote that you use a Random-Effects meta-analysis. However, the figures show that often the fixed-effects model was used. Probably only forgotten to change in RevMan. Before we can send the article to peer review this should be revised."

Response: Thank you for noticing this error. You are quite right that this was a RevMan issue. This has now been corrected: All figures have been amended accordingly, and numbers reported in the text updated where necessary.

---

## [Decision Letter · Decision Letter 1]

16 Feb 2021

PONE-D-20-35048R1

Self-management for chronic widespread pain including fibromyalgia: A systematic review and meta-analysis

PLOS ONE

Dear Dr. Geraghty,

Thank you for submitting your manuscript to PLOS ONE. After careful consideration, we feel that it has merit but does not fully meet PLOS ONE’s publication criteria as it currently stands. Therefore, we invite you to submit a revised version of the manuscript that addresses the points raised during the review process.

We look forward to receiving your revised manuscript.

Kind regards,

Tim Mathes

Academic Editor

PLOS ONE

Reviewers' comments:

Reviewer's Responses to Questions

**Comments to the Author**

1. If the authors have adequately addressed your comments raised in a previous round of review and you feel that this manuscript is now acceptable for publication, you may indicate that here to bypass the “Comments to the Author” section, enter your conflict of interest statement in the “Confidential to Editor” section, and submit your "Accept" recommendation.

Reviewer #1: All comments have been addressed

Reviewer #2: (No Response)

Reviewer #3: All comments have been addressed

2. Is the manuscript technically sound, and do the data support the conclusions?

Reviewer #1: Yes

Reviewer #2: Yes

Reviewer #3: Yes

3. Has the statistical analysis been performed appropriately and rigorously? 

Reviewer #1: No

Reviewer #2: Yes

Reviewer #3: Yes

4. Have the authors made all data underlying the findings in their manuscript fully available?

Reviewer #1: Yes

Reviewer #2: Yes

Reviewer #3: Yes

5. Is the manuscript presented in an intelligible fashion and written in standard English?

Reviewer #1: Yes

Reviewer #2: Yes

Reviewer #3: Yes

6. Review Comments to the Author

Reviewer #1: Thanks for the invitation for reviewing the paper. I think this revised version is generally good in methodology, and the results and conclusions are also important for the application of self-management in the management of chronic widespread pain including fibromyalgia.

I only suggest that it would be better in methodology to assess the grade of evidence body (the pooled effect size of meta analysis) by using GRADE (Gradings of Recommendations Assessment, Development, and Evaluation).

Reviewer #2: Thank you for the opportunity to review this paper. It is a well written and comprehensive systematic review and meta-analysis of self-management interventions used to manage chronic pain, including Fibromyalgia.

I have a few minor comments for the authors to consider.

1. Lines 177 – it is not clear who first conducted the initial searches. Was this by the research team members or by the librarian?

2. Lines 173-177 are about the search are repetitive.

3. Lines 756- 758. The authors make an interesting point that further research should be conducted in primary care. Therefore, could the authors make it clearer in the delivery modality section where the SM interventions were delivered? For example, the authors highlight that the studies were outpatient-based (line 276) could the authors clarify whether the studies in their review were conducted in a hospital or community outpatient setting?

Reviewer #3: Thank you for the opportunity to review this interesting, well-written, and timely systematic review and meta-analysis on self-management interventions for CWP/Fibromyalgia.

In my opinion, this review is excellent, but could be improved in a few, minor ways:

1. It would be helpful if the authors define short- and long-term in the abstract.

2. It would be helpful if the authors define what pain construct they are referring to in the abstract. Pain has many constructs and its only after reading the paper that the reader understands the focus is on pain intensity/severity rather than another construct.

3. I would recommend that the authors proofread the entire review. I found several small editing errors that can distract from an important message of the review. Examples: page 6, line 152; page 10, line 222; page 11, line 227; page 11, line 229; page 13, line 294; page 21, line 375; page 22, line 393 (awkward wording); page 23, line 425, page 31, line 608;

4. Unclear sentence: Page 13, lines 283-285. Please clarify.

7. PLOS authors have the option to publish the peer review history of their article (what does this mean?). If published, this will include your full peer review and any attached files.

Reviewer #1: **Yes: **Shizheng Du

Reviewer #2: No

Reviewer #3: No

---

## [Author Response · Author response to Decision Letter 1]

25 Jun 2021

Response to reviewers

Reviewer #1: 

Thanks for the invitation for reviewing the paper. I think this revised version is generally good in methodology, and the results and conclusions are also important for the application of self-management in the management of chronic widespread pain including fibromyalgia.

Response: We thank the reviewer for their positive comments. 

I only suggest that it would be better in methodology to assess the grade of evidence body (the pooled effect size of meta-analysis) by using GRADE (Gradings of Recommendations Assessment, Development, and Evaluation).

Response: We agree, the use of GRADE would improve our methodology. We have now employed the GRADE framework to rate the quality of evidence for our primary outcomes (physical function and pain) and our two comparisons: Self-management vs. usual care/no treatment controls and self-management vs. active comparison. 

Importantly, in the present review we included studies that both could and could not be meta-analysed. We did this to avoid bias that may occur when review teams only include trials with data suitable for meta-analysis. Murad et al. provide guidance on how to include studies without estimates of effect into the GRADE framework. We followed this approach to assess the quality of evidence that we summarised narratively, as well as our meta-analyses. We now include summary of findings tables (Table 2 and Table 3) on pages 24 to 27, along with full evidence profiles as supporting information (S4 and S5). We have also included the following text in the methods section on pages 10 and 11:

“We used the GRADE (Grading of Recommendations Assessment, Development, and Evaluation) approach to rate the quality of evidence in the review for our primary outcomes (29). When using GRADE, evidence on outcomes from RCTs starts as high quality, and reviewers then rate down for limitations (risk of bias), inconsistency, indirectness, imprecision, and publication bias (ratings range from ‘high’, to ‘very low’). Evidence for outcomes can be rated up considering factors such as very large effects and evidence of dose response gradients (29). Whilst this approach is frequently coupled with meta-analysis, Murad et al. (30) show how it can also be used in absence of a quantitative estimate of effect. As such, we included both meta-analysed and narratively reviewed studies when grading the quality of evidence for the primary outcomes. Where estimates and confidence intervals were absent in narratively reviewed studies, we took a cautionary approach and rated down for imprecision. We prepared evidenced profiles and summary of findings tables for our two comparisons: self-management vs. usual care/no treatment, and self-management vs. active comparison. GRADE ratings were agreed though consensus by a sub-team comprising AG, BS, CP and EM”

The following text has been included in the results section on page 23: 

“The quality of the evidence for the review outcomes was rated as low in most cases. Table 2 and Table 3 show our summary of findings for our two main comparisons (self-management vs. usual care and self-management vs. active comparison). Most outcomes were rated down with a combination of serious limitations (risk of bias) and either serious inconsistency, or serious imprecision. Further details can be found in the full evidence profiles provided as supporting information (S4 and S5).

Reviewer #2: Thank you for the opportunity to review this paper. It is a well written and comprehensive systematic review and meta-analysis of self-management interventions used to manage chronic pain, including Fibromyalgia.

Response: We are grateful for the positive comments on our review. 

I have a few minor comments for the authors to consider.

1. Lines 177 – it is not clear who first conducted the initial searches. Was this by the research team members or by the librarian?

Response: We apologise for this lack of clarity. We have now clarified that systematic review specialist, EM, conducted all searches. On page 6, line 126 and 127 we have added the following:

“All searches were conducted by systematic review specialist, EM.”

2. Lines 173-177 are about the search are repetitive.

Response: We are not clear on the source of the repetition mentioned. These lines refer to the screening and selection of appropriate titles and abstracts, then the further screening and selection of full text papers. Although similar, both are important parts of the review process. Thus, we have left this section for now. We are happy to amend further following editorial guidance if necessary. 

3. Lines 756- 758. The authors make an interesting point that further research should be conducted in primary care. Therefore, could the authors make it clearer in the delivery modality section where the SM interventions were delivered? For example, the authors highlight that the studies were outpatient-based (line 276) could the authors clarify whether the studies in their review were conducted in a hospital or community outpatient setting?

Response: Unfortunately, the included studies rarely provided detail on the nature of the outpatient setting (where the groups took place). We have now clarified this in the Table of Characteristics (S3 in supporting information), noting that the setting location is not clear in the majority of cases. A small number of studies did list ‘leisure centre’ or a ‘clinic’ with regard to location, which we have now added to the Table of Characteristics. 

To include further relevant details related to this reviewer’s helpful point regarding primary care, we now include details of where participants were recruited from. For example, from primary care, hospital rheumatology clinics or community advertisements. We include the following in the ‘Participants’ section on page 13:

“Broadly, 46.2% of participants were recruited from rheumatology clinics or hospital specialist settings; 12.8% were recruited from primary care; 15.4% were recruited from a mixture of primary care and rheumatology clinics; 12.8% were recruited from community advertisements, and in a further 12.8% of cases it was not clear where participants were recruited from.”

Reviewer #3: Thank you for the opportunity to review this interesting, well-written, and timely systematic review and meta-analysis on self-management interventions for CWP/Fibromyalgia.

In my opinion, this review is excellent, but could be improved in a few, minor ways.

Response: We appreciate this reviewer’s positive comments. 

1. It would be helpful if the authors define short- and long-term in the abstract.

Response: We have now done this where short and long term are first used in the abstract on page 2:

“Despite some variability in studies narratively reviewed, in studies meta-analysed self-management interventions improved physical function in the short-term, post-treatment to 3 months (SMD 0.42, 95% CI 0.20, 0.64) and long-term, post 6 months (SMD 0.36, 95% CI 0.20, 0.53), compared to no treatment/usual care controls.”

2. It would be helpful if the authors define what pain construct they are referring to in the abstract. Pain has many constructs and its only after reading the paper that the reader understands the focus is on pain intensity/severity rather than another construct.

Response: We agree, this is a helpful addition. We now refer to pain intensity in the methods section in the abstract when listing our primary outcomes, on page 2:

“Primary outcomes included physical function and pain intensity.”

In addition, this useful point regarding constructs led to us reflecting on physical function, our other primary outcome. We have now provided further detail of how physical function was measured in the studies on pages 28-29:

“Studies that presented subjective physical function outcomes used a range of measures. The most commonly used included the Fibromyalgia Impact Questionnaire (FIQ) physical functioning subscale [48, 49, 52, 54, 58, 71], and the physical functioning item from the SF-36 [46, 68]. Some studies used the SF-36/8 physical component summary as a primary measure of physical function [32, 55]. Consequently, for consistently, where study authors did not present alternative physical functioning measures, but did present an SF-36/12/8 physical component summary score this was used as a measure of physical functioning.” 

3. I would recommend that the authors proofread the entire review. I found several small editing errors that can distract from an important message of the review. Examples: page 6, line 152; page 10, line 222; page 11, line 227; page 11, line 229; page 13, line 294; page 21, line 375; page 22, line 393 (awkward wording); page 23, line 425, page 31, line 608;

4. Unclear sentence: Page 13, lines 283-285. Please clarify.

Response: We apologise for these editing errors. We have now carefully worked through the manuscript and corrected or clarified these issues. 

Additional minor amendments:

In addition to the above amendments, as part of the revision process we undertook a further full accuracy check on all data included. As part of this process, we noted some minor corrections were needed to the study sample sizes in the meta-analyses. These corrections were made, and all relevant meta-analyses were rerun. All relevant figures have been updated. 

We also noted that two studies needed to come out of the FIQ long term analysis; on further consideration we felt they did not meet our definition of long-term. This reduced the number included in the analysis to below the number required for funnel plot asymmetry and applying Egger’s test.

We had included McBeth et al.’s 2012 results on physical function in our narrative section on self-management interventions vs. active comparisons. However, on review as part of the revision process, we added the quantitative data from this study on physical function to the relevant meta-analysis. McBeth et al.’s measure of chronic pain grading has also been added to the relevant section. 

We have tracked all amendments for ease of review. The above additions do not change the original interpretation of the review.

---

## [Editor Report · Decision Letter 2]

1 Jul 2021

Self-management for chronic widespread pain including fibromyalgia: A systematic review and meta-analysis

PONE-D-20-35048R2

Dear Dr. Geraghty,

We’re pleased to inform you that your manuscript has been judged scientifically suitable for publication and will be formally accepted for publication once it meets all outstanding technical requirements.

Kind regards,

Tim Mathes

Academic Editor

PLOS ONE
---

## [Editor Report · Acceptance letter]

7 Jul 2021

PONE-D-20-35048R2 

Self-management for chronic widespread pain including fibromyalgia: A systematic review and meta-analysis 

Dear Dr. Geraghty:

I'm pleased to inform you that your manuscript has been deemed suitable for publication in PLOS ONE. Congratulations! Your manuscript is now with our production department. 

Kind regards, 

on behalf of

Dr. Tim Mathes 

Academic Editor

PLOS ONE